# The operational environment and rotational acceleration of asteroid (101955) Bennu from OSIRIS-REx observations

C.W. Hergenrother[1], C.K. Maleszewski[1], M.C. Nolan[1], J.-Y. Li[2], C.Y. Drouet d'Aubigny[1], F.C. Shelly[1], E.S. Howell[1], T.R. Kareta[1], M.R.M. Izawa[3], M.A. Barucci[4], E.B. Bierhaus[5], H. Campins[6], S.R. Chesley[7], B.E. Clark[8], E.J. Christensen[1], D.N. DellaGiustina[1], S. Fornasier[4], D.R. Golish[1], C.M. Hartzell[9], B. Rizk[1], D.J. Scheeres[10], P.H. Smith[1], X.-D. Zou[2] & D.S. Lauretta[1] & The OSIRIS-REx Team

During its approach to asteroid (101955) Bennu, NASA's Origins, Spectral Interpretation, Resource Identification, and Security-Regolith Explorer (OSIRIS-REx) spacecraft surveyed Bennu's immediate environment, photometric properties, and rotation state. Discovery of a dusty environment, a natural satellite, or unexpected asteroid characteristics would have had consequences for the mission's safety and observation strategy. Here we show that spacecraft observations during this period were highly sensitive to satellites (sub-meter scale) but reveal none, although later navigational images indicate that further investigation is needed. We constrain average dust production in September 2018 from Bennu's surface to an upper limit of $150 \, g \, s^{-1}$ averaged over 34 min. Bennu's disk-integrated photometric phase function validates measurements from the pre-encounter astronomical campaign. We demonstrate that Bennu's rotation rate is accelerating continuously at $3.63 \pm 0.52 \times 10^{-6}$ degrees $day^{-2}$, likely due to the Yarkovsky–O'Keefe–Radzievskii–Paddack (YORP) effect, with evolutionary implications.

[1] Lunar and Planetary Laboratory, University of Arizona, Tucson, AZ, USA. [2] Planetary Science Institute, Tucson, AZ, USA. [3] Institute for Planetary Materials, Okayama University-Misasa, Misasa, Tottori, Japan. [4] LESIA, Observatoire de Paris, Université PSL, CNRS, Sorbonne Université, Univ. Paris Diderot, Sorbonne Paris Cité, Meudon, France. [5] Lockheed Martin Space, Littleton, CO, USA. [6] Department of Physics, University of Central Florida, Orlando, FL, USA. [7] Jet Propulsion Laboratory, California Institute of Technology, Pasadena, CA, USA. [8] Department of Physics and Astronomy, Ithaca College, Ithaca, NY, USA. [9] Department of Aerospace Engineering, University of Maryland, College Park, MD, USA. [10] Smead Department of Aerospace Engineering, University of Colorado, Boulder, CO, USA. A full list of authors and their affiliations appears at the end of the paper. Correspondence and requests for materials should be addressed to C.W.H. (email: chergen@orex.lpl.arizona.edu).

The Approach phase for the OSIRIS-REx mission occurred between 17 August 2018 and 2 December 2018[1]. Observations of near-Earth asteroid (NEA) Bennu began as the asteroid was just bright enough for detection by the OSIRIS-REx Camera Suite (OCAMS)[2]. The three goals for the Approach phase of the mission were to optically acquire the asteroid, survey the vicinity of the asteroid for any hazards (natural satellites or dust trails) that may be present, and characterize the asteroid point-source properties for comparison with ground- and space-based telescopic data[1,3] Here we show that we detect no hazards within the sub-meter sensitivity limits of our Approach phase observations. Further work will follow since images from our navigation camera later revealed the existence of apparent particles in close vicinity of Bennu[4]. We find strong agreement between the pre-encounter and OSIRIS-REx disk-integrated photometric properties. We detect a continuous acceleration in Bennu's rotation and conclude that it results from the YORP effect[5].

## Results

**Search for dust**. The first dedicated science observation of Bennu by OSIRIS-REx was a search for dust on 11 and 12 September 2018 when Bennu was at a heliocentric distance of 1.21 au. We used the OCAMS PolyCam and MapCam instruments[2] to survey all space within a 35,000 km radius of Bennu. If present, unbound dust released due to outgassing processes would appear as diffuse cometary features (trails or tails) along the directions between the anti-solar and anti-heliocentric velocity vectors[6]. The shape of such features is governed by the balance between radiation pressure and gravity and is particle size-dependent. Only particles ejected in the anti-solar direction two to eight weeks prior to the observations would have been detected in our MapCam images. We observed no detectable dust during this search.

We determined upper limits of 300 kg for the dust mass within 405 km of Bennu and 150 g s$^{-1}$ averaged over 34 min for the average dust production rate by considering the properties and ephemeris of Bennu, assuming a dust flux such as that from a near-surface coma, and assuming the dust ejection velocity measured for comet 67 P at perihelion (see methods). Observations in the thermal infrared taken with the Spitzer space telescope (hereafter, Spitzer) yielded upper limits on dust mass from Bennu of 1000 kg and on dust production rate of 1 g s$^{-1}$ (ref. [7]). The OSIRIS-REx Approach phase dust searches were comparable in sensitivity to Spitzer observations with regards to dust mass but much less sensitive to lower dust production rates.

Our upper limit on Bennu's average dust production rate is on the extreme lower end of those of main belt comets, which are in the range of 100 to 4000 g s$^{-1}$ (ref. [8]). Asteroid (3200) Phaethon, the parent of the Geminid meteor shower, reaches a peak dust production of approximately 300 g s$^{-1}$, (ref. [9]) during its perihelion passage, possibly driven by thermal desiccation of surface rocks. As Bennu's surface has many rocks with apparent fracturing that may have been thermally induced[10,11] similar dust production due to thermal desiccation may occur on Bennu.

We will conduct two dedicated searches for dust mass loss at a high phase angle of approximately 130 degrees during the Detailed Survey phase of the mission in spring 2019[1]. The higher phase angle of these searches enhances dust observability if the dust particles are forward-scattering, as would be expected for sub-micron-sized particles[12].

**Search for natural satellites**. The fraction of NEAs larger than 300 m with satellites is 15 ± 4%[13]. The smallest asteroid satellite yet observed is approximately 44 m in diameter, around the primary body NEA 2000 CO$_{101}$[14]. Pre-encounter modeling suggests that any Bennu satellites with diameters larger than 1 m

could be stable on orbits within 20 km, and diameters of 10 cm could be stable on orbits within 12 km[15,16] Ground-based radar observations were sensitive to satellites of Bennu down to a size of 2 to 20 m diameter, depending on the rotation rate of the satellites[17].

For OSIRIS-REx, we calculated the detectable satellite size using the lower-limit bound of Bennu's albedo (0.03) and the upper-limit on steepness of the phase function slope (0.043 mag per degree) determined from the ground[18]. We tested different exposure durations to decrease the glare of Bennu and allow detections down to a projected height of 20 m above the sunlit limb. From the multiple search dates, the cadence of images taken per date, and multiple search methods—including visual inspection and use of the asteroid-hunting Catalina Sky Survey automated moving object detection software[19]—we estimated that our detection efficiency would be approximately 99% for satellites 10 cm and larger. Orbiting satellites would spend some fraction of their orbits in front of or behind the asteroid. The multiple dates and 5-h observing windows per date ensured that orbiting objects were detectable.

We conducted our search for natural satellites using PolyCam (Fig. 1). Initially, when the spacecraft was at a range of 3100 km, the search was over an area of 60 km radius when our observations were sensitive to 0.5 m or larger satellites. Later, as the spacecraft got closer, satellites as small as 8 cm would have been bright enough to be imaged within 18 km of Bennu. As the spacecraft approached Bennu, the PolyCam field of view narrowed down to smaller areas of the sky, allowing us to search for smaller satellites (Fig. 2). In all, we used PolyCam and MapCam on ten dates to search for satellites, and our observations were sensitive to objects as small as 24 cm in diameter within the entire Hill Sphere (31 km)[20] (Supplementary Table 1).

The Hayabusa mission could detect satellites down to 1 m around NEA (25143) Itokawa[21], the Rosetta mission could detect satellites between 1 and 6 m around comet 67 P[22], and the Dawn mission could detect satellites as small as 3 m around (4) Vesta and 12 m around (1) Ceres[23,24]. In contrast, the high sensitivity of our OCAMS instruments allowed us to search for satellites at the sub-meter scale. No satellites were detected down to our sensitivity limits.

Although we detected no dust or satellites during our dedicated search with the OCAMS instruments during the Approach phase, images collected in January 2019 by one of the navigation cameras (NavCam1)[25] have since revealed apparent particles in the vicinity of Bennu[4]. These probable particles are likely to be smaller than the sensitivity limit of the Approach-phase satellite search. Their nature and production mechanism are still under investigation, although they appear to originate from Bennu. We plan to continue monitoring the near-Bennu environment throughout the rest of the mission.

**Disk-integrated phase function**. From the ground, the disk-integrated phase function of Bennu was determined using measurements obtained between 2005 and 2012[18]. For direct comparison to the ground-based data, we first limited the OCAMS data to the same phase angle range as the ground-based data. This analysis yields a linear fit with $H_v = 20.51 ± 0.04$ and $B_v = 0.039 ± 0.001$, in good agreement with the ground-based data. The phase slope of Bennu is similar to that of other low-albedo main belt asteroids such as the C, F, and P types (0.04 to 0.05 mag per degree of phase angle)[26].

We then modeled the full disk-integrated phase dependence of Bennu with several methods, including the Lommel-Seeliger model[27], the IAU H-G model[28], and the more recently adopted

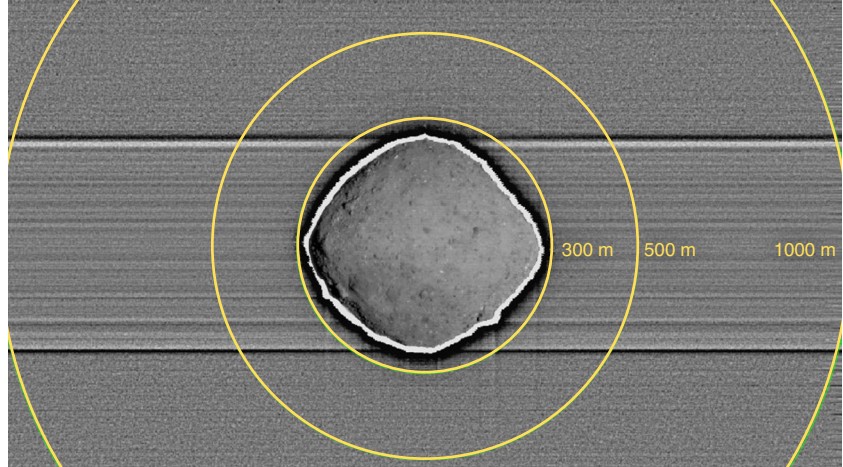

**Fig. 1** Satellite-search imaging of the near-asteroid region. The image is a median combination of 15 PolyCam exposures from 10 November 2018 tracking the motion of the asteroid. This method enhances the relative detectability of satellites while suppressing the signal of stars and other background astronomical objects. The yellow circles denote distances from the Bennu center of figure. The horizontal features are due to charge smear. No satellites are apparent

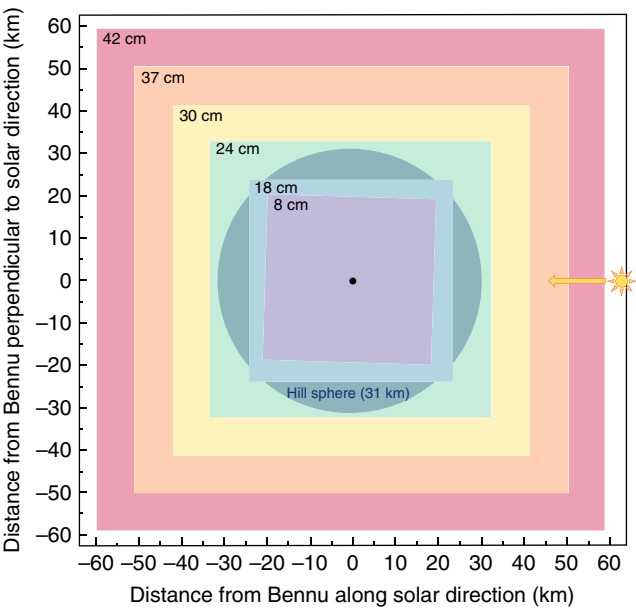

**Fig. 2** Schematic illustrating visibility of detectable natural satellites. The minimum detectable diameter is given within the upper left corner of each search region. Bennu is marked at the center of the figure. Also marked is the Hill Sphere, or the region in which objects remain in orbit due to Bennu's gravity. We completed an extensive search for natural satellites that sampled the entire Hill Sphere with a sensitivity to 24 cm satellite diameter. We detected none at this sensitivity level. Source data are provided in Supplementary Table 1

$H,G_1,G_2$ and $H,G_{12}$ models[29,30] (see Methods) (Fig. 3). For each model, the parameter $H$ is the absolute magnitude. Each particular value of $G$ helps define the curve shapes of each model. The IAU $H$-$G$ model fails to correctly reproduce the data at very small phase angles, as it predicts a larger opposition effect than is present. Difficulty modeling the opposition effect of low-albedo objects is a recognized deficiency of this model[31]. The $H,G_1,G_2$ model results in an artificial drop in the magnitude at phase angles < 1°. We fitted the v-band phase function data of Bennu with both the original and the revised $H,G_{12}$ models[29,30] (Supplementary Table 2). The revised $H,G_{12}$ model fits the data

in the full range of phase angles that they cover (Fig. 3). In general, the $G$-parameters ($G$, $G_1$, $G_2$, and $G_{12}$) of all three models suggest a shallower phase slope with increasing wavelength, consistent with phase reddening in the visible wavelengths[10].

The disk-integrated Lommel-Seeliger model yields a geometric albedo of $0.044 \pm 0.002$ and Bond albedo of 0.0170. The $H,G_{12}$ model found an $H_v$ of $20.41 \pm 0.02$. This compares well with the best pre-encounter $H_v$ value of $20.56^{+0.05}_{-0.15}$[18]. The ground-based photometry did not include observations at phase angles less than 15 degrees phase angle but did extend up to 100 degrees phase angle. The good agreement between the ground-based and OSIRIS-REx photometric results highlights the importance of supporting ground-based asteroid observations at multiple observing geometries.

Comparing the magnitude derived from the linear fit (i.e. excluding the opposition effect) with the absolute magnitude from the revised $H,G_{12}$ model, we found a magnitude increase ($\Delta m$) of 0.20 mag that could be related to the opposition effect (Fig. 3, inset). The parameters of Bennu's opposition effect are consistent with the values reported in the literature for about 40% of low-albedo asteroids with average opposition effect amplitudes of $0.16 \pm 0.05$ mag and starting phase angles of 6 to 7 degrees[32,33].

**Rotation state**. We had extensive knowledge of the rotation state of Bennu from our pre-encounter astronomical campaign[3,17,18] when lightcurve observations yielded a synodic rotation period of $4.2905 \pm 0.0065$ h[18]. The observed low amplitude and trimodal (three maxima and three minima) lightcurve was consistent with the rotation of a nearly spherical body observed at high phase angles. We used the best-fit shape and pole position for Bennu from a combination of radar images and lightcurve data[17]. The sidereal rotation period determined from the lightcurve and radar data was 4.29746 ± 0.002 h. Bennu's obliquity was determined to be $178 \pm 4$ degrees with the rotation pole at $(87, -65) \pm 4$ degrees (J2000 equatorial coordinates)[17]. From ground-based and Hubble Space Telescope observations in 1999, 2005, and 2012, an increase was detected in the rotation rate of Bennu of $2.64 \pm 1.05 \times 10^{-6}$ degrees day$^{-2}$, possibly due to YORP thermal torques[34].

For the present study, we measured Bennu's rotation rate by obtaining a series of asteroid light curves using the OCAMS MapCam instrument with the b′, v, w, and x spectral filters (centered at 470, 550, 770, and 860 nm wavelengths, respectively)[1].

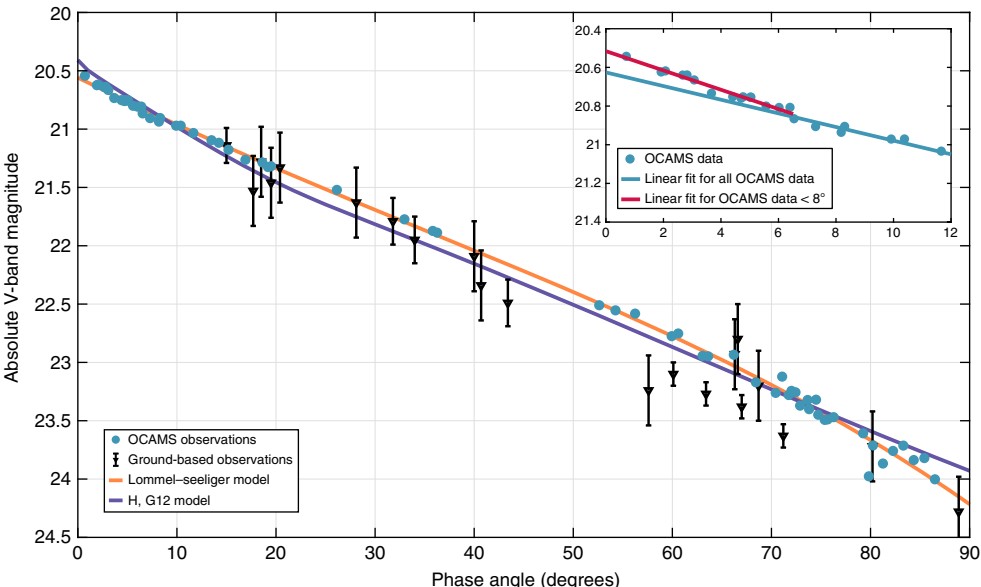

**Fig. 3** Phase function observations and models for Bennu. Over-plotted are the model fits to the new data: a Lommel-Seeliger model and a revised $H,G_{12}$ model. The ground-based measurements are published in[18]. We originally converted the ground-based measurements to the v-band magnitude scale, plotted with 1-sigma photometric error bars. The 1-sigma photometric error bars for OCAMS data are less than or equal to the size of the data points. Inset, upper right: The OSIRIS-REx measurements at low phase angles (0 to 12 degrees). We plot this subset of the data to illustrate the small opposition effect present for asteroid Bennu. The small opposition effect is consistent with what we would expect for low-albedo, carbonaceous bodies. The red line is the linear fit to observations made at less than 7.5 degrees phase angle. The blue line is the linear fit to all OSIRIS-REx data points at greater than 7.5 degrees phase angle. Source data are provided in Supplementary Tables 2 and 3

The OSIRIS-REx lightcurves have a different shape than the ground-based lightcurves, with four peaks instead of three, due to the Approach phase data being taken at much lower solar phase angle (4 to 18 degrees) than any of the earlier telescopic data (60 to 71 degrees).

Bennu's rotation rate has accelerated since 1999, but previous work could not distinguish between continuous spin-up due to YORP and a step change in rotation rate due to a change in moment of inertia[34]. Here we confirm that the acceleration has continued to the present day and is most likely due to the YORP effect. We derive an acceleration of $3.63 \pm 0.52 \times 10{-6}$ degrees day$-2$, with a rotation rate of $2011.1697 \pm 0.0011$ degrees day$-1$, and a period of $4.296007 \pm 0.000002$ h (compared to a rotation rate of $2011.1445 \pm 0.0011$ degrees day$-1$ and period of $4.296061 \pm 0.000002$ h at the J2000 epoch). Figure 4 shows the OSIRIS-REx Approach phase lightcurve and model fits using the shape model and this acceleration (black curve). The acceleration that we determine is a better fit to the observations than the previously determined value[34] (Fig. 4, orange curve) or a constant rotation rate since 2009 (Fig. 4, green curve). The inset plot shows that the rotational phase determined at four epochs is consistent with a rotation rate increasing along a quadratic curve. This is evidence for a continuous change—i.e., YORP—not a step change.

The doubling time for this acceleration is about 1.5 million years, indicating that Bennu's surface could be unstable on million-year timescales. However, such timescales are much shorter than Bennu's apparent surface age of at least 100 million years[10,20] suggesting that the YORP acceleration has changed over time, possibly due to changes in Bennu's orbit or its shape[35]. All of the NEAs, including Bennu, for which YORP has been detected have been accelerating[36], and many of the top-shaped NEAs are binary systems; perhaps Bennu will evolve to this state. The OSIRIS-REx spacecraft will continue to probe the near-surface environment of Bennu to resolve the relationship between its surface and its rotation state.

## Methods

**Search for dust near Bennu.** A sequence of images centered on Bennu was obtained using the OCAMS PolyCam and MapCam imagers. The image sequence was median co-added on the motion of Bennu to produce a map of the dust near Bennu. Dust would have exhibited itself as diffuse features either around Bennu, trailing Bennu in the anti-solar direction, or trailing Bennu along its orbit.

PolyCam dust plume search images were collected on 11 September 2018 when Bennu was at a range of 1.05 million km and phase angle of 43 degrees. The MapCam images were collected on the following date when Bennu was at a range of 1.00 million km and phase angle of 44 degrees. On these dates, PolyCam covered a region of Bennu's orbit extending 7300 km leading and trailing Bennu. MapCam covered a region along Bennu's orbit that extends 35,000 km leading and trailing Bennu.

The 11 and 12 September dates were chosen because Bennu's apparent position in J2000 celestial coordinates placed it in a part of the Milky Way that is relatively less dense in stars. From the beginning of the Approach phase to the time of the spacecraft's second asteroid approach maneuver (15 October), Bennu traversed a very dense part of the Milky Way as seen from the spacecraft. Although none of the dates were optimal, 11 and 12 September were the best available while also maximizing the region of space around Bennu that was searched for dust.

**Modeling of possible dust trails for Bennu.** The model of ref. [6] was implemented to model potential dust trails from Bennu. We use an adapted version of the code originally developed by Jean-Baptiste Vincent[37]. The Vincent version of the model uses numerical integration to track the position and velocity of a dust particle that is ejected from the surface of the parent body. This is opposed to the analytical equations first proposed in ref. [6]. Most of the adaptations we made involve changing the location of observing from Earth to the spacecraft's position. This allowed us to generate trail locations as a function of right ascension and declination.

For simplicity, we assumed constant particle sizes as a function of beta as opposed to implementing a particle size distribution found in other versions of this model A particle size for a particular syndyne can be calculated from Eq. 3 in ref. [6]. To estimate times in which the earliest particles ejected, trails from our model were plotted over the median co-added images of Bennu from the hazards search. The combined plot was visually inspected to determine at what time each trail would leave the field-of-view of the co-added image. The range of particle sizes used to estimate ejection time are for beta values of 0.01 and 0.1. This corresponds to grain sizes of 66.1 and 3.3 microns respectively.

**Determining the bound on the mass loss rate.** To provide a bound on the mass loss rate of a detectable coma, we adapt a method[38] used for members of the

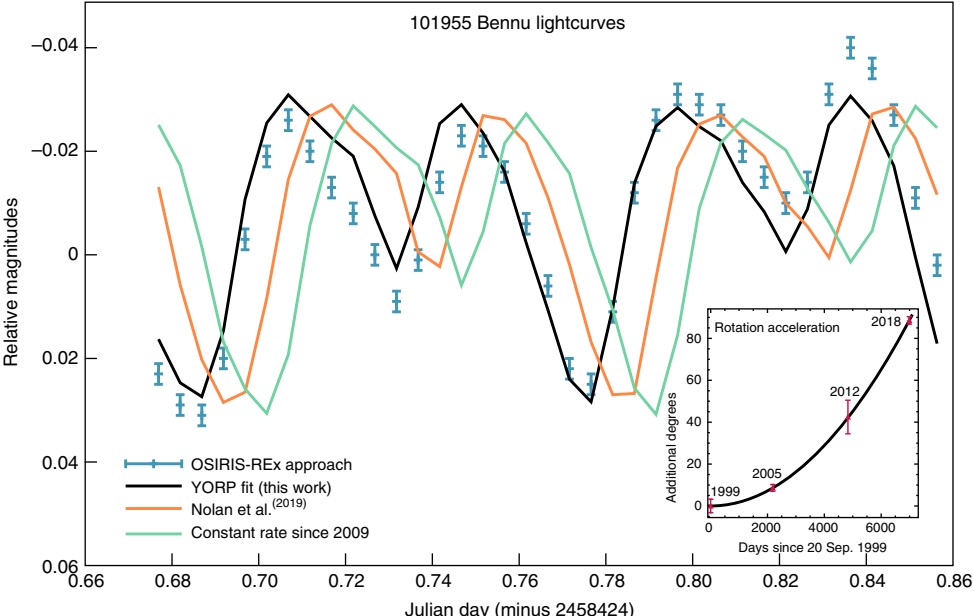

**Fig. 4** Lightcurve data and models for Bennu on 2 November 2018. The blue crosses are the OSIRIS-REx observations with their associated 1-sigma photometric uncertainties. The black curve (best fit to peaks and minima) shows the YORP acceleration determined from fitting the Approach phase observations with 1-sigma uncertainties. The orange lightcurve using the previously reported acceleration value[34] does not fit as well as the black curve. The green curve assumes a constant rotation rate since 2009. The inset plot shows the quadratic increase of rotation phase with time, consistent with rotational acceleration due to YORP. Source data are provided as a Source Data file

Centaur population. The goal of this method is to estimate the total mass of a possible coma from light measured using photometry of an annulus between two circular apertures. We also estimate the approximate time that dust would remain in the given annulus. Dividing these two quantities provides us with a mass loss rate.

A median co-add of images from the Dust Search campaign was created, and then two photometry measurements of Bennu at radii of 20 and 30 arcseconds were measured. The IRAF phot package was used to determine instrumental magnitude of these apertures. We also corrected for sections of the point spread function leaking into the coma annulus as suggested in ref. [38]. These magnitudes were transformed into a R-band magnitude system where the apparent magnitude of the Sun is known (for Eq. 3).

Equations 1, 3, and A4 in ref. [38] were used to calculate the bound mass in the annulus between the two photometric apertures. Phase angle, Bennu-Sun distance, and Bennu-OSIRIS-REx distances were calculated from Bennu's ephemeris using JPL HORIZONS. The phase darkening correction was interpolated from the phase darkening curve developed in refs. [12,39]. To convert from cross-section to mass (equation A4), we assume that the R-band albedo is equal to the V-band albedo and set it to a value of 0.045. We assume the same particle sizes, $a_+ = 1$ cm and $a_- = 0.1$ micron[38,40].

The residence time is calculated from Eq. 4 in ref. [38]. We adopt an outflow velocity of 25 m s$^{-1}$, which is the estimated value of the ejection velocity for 67P/Churyumov-Gerasimenko at perihelion[41]. This value is chosen since the perihelion of 67P is close to the Bennu-Sun distance during the hazards search (1.3 vs. 1.2 au). It also agrees with measurements taken of the ejection velocity from the Rosetta mission[42].

**Search for natural satellites around Bennu**. Dedicated searches for natural satellites in orbit around Bennu were conducted on 10 separate dates. A summary of the observing circumstances and detection limits of each date is given in Supplementary Table 1. Observations were collected over the course of 5 h (between 4:00 and 9:00 UTC) on each date.

Each date of the search consisted of 3 × 3 mosaic fields with approximately 10% overlap between each field. Each field was imaged between 15 and 30 times over a span of time allowing an object with Bennu rates of motion to move between 5 and 30 pixels relative to the background stars. This dwell time on each field ranged from 4.2 to 25.7 min depending on Bennu's varying rates of relative motion. Each field was visited between 2 and 4 times on each date. As a result, each field was imaged 60 times per date. Exposure times ranged from 5 s to 0.15 s. Shorter exposure times were used during the later search dates to avoid saturation and pixel blooming near the asteroid. The shorter exposure times were set to still allow the detection of satellites as small as 10 cm.

On 23 to 28 October 2018, only PolyCam was used. Observations between 30 October and 11 November 2018 used a combination of MapCam and PolyCam.

Each post–30 October search consisted of a 3 × 3 MapCam mosaic and a single PolyCam field centered on Bennu. The 30 and 31 October PolyCam fields were not used for satellite searching as Bennu was located near the edge or outside the small PolyCam field-of-view due to the greater navigational uncertainty after the third asteroid approach maneuver.

Three different methods were utilized to search for satellites within the OCAMS images. The first involved manually blinking the 15 or 30 images taken per field per visit for moving objects. The second combined, or stack and shifted, all of the images taken during a visit on the motion of Bennu. The combined images minimized the signal of background stars while enhancing the brightness of objects moving at the rate of Bennu. The third search method used the moving object detection software of the Catalina Sky Survey (CSS) to automatically detect satellites[19]. The CSS software was used on 5 images at a time. Due to the large number (15–30) images taken per visit, up to 6 different combinations of the images were run through the CSS software enabling multiple opportunities for detection. The software approach was not used on 23–25 October due to the slow apparent motion of Bennu and any satellites relative to the background stars. It was also not applicable on 30–31 October and 10–11 November due to the short exposure times used and the low number of detectable background stars.

The sensitivity and efficiency of the satellite search was improved by conducting a search for Earth Trojan Asteroids (ETAs) during the outbound cruise phase in February 2017[43,44] By exercising the entire moving object detection process, lessons learned during the ETAs search resulted in changes to detection software, number and cadence of observations, exposure times, and the use of both PolyCam and MapCam.

**OCAMS disk-integrated photometry calibration**. The combination of the OCAMS narrow point spread function (PSF) with its detector's strongly non-uniform pixel response makes photometric calibration using standard stars more challenging than expected. A dedicated calibration campaign has yielded valuable insights but disappointing results. A second campaign incorporating lessons learned from our first attempt is planned. In the interim, we use defocused images of open star cluster NGC 3532 to derive an absolute radiometric calibration for the PolyCam. We then use near simultaneous MapCam and PolyCam observations of Bennu to transfer this calibration to the MapCam.

During outbound cruise, PolyCam acquired a through-focus sequence of images of NGC 3532. In one of these images, stars are defocused enough to cover approximately 100 pixels, thereby minimizing the effects of aliasing. We exclude stars near the edge of the detector or in pixel regions which do not behave like the bulk of the detector. Stars for which any raw counts are out of the linear range for PolyCam (<12,500 DN) are also rejected. Despite the significant defocus, stars are still well resolved and can be automatically detected, identified and measured. Stars close to each other are also excluded. Visual inspection of each of the remaining stars is used to exclude a further 95 stars with PSF indicative of one or more

unresolved companions, leaving 187 stars in our sample. In addition to being an open cluster, NGC 3532 is also a diffuse nebula, so we estimate and remove the local background of the nebula at each star.

Given the panchromatic filter's 650 nm center wavelength, the integrated star flux is then compared to the R magnitude reported by the American Association of Variable Star Observers (AAVSO) NGC 3532 Standard Field catalog. The fit between the logarithm of measured DN/s and the AAVSO Catalog $m_R$ magnitudes is very good (R = 0.998). When corrected to the OSIRIS-REx reference temperature that fit is:

$$m_R = -2.5 Log10(DN/s(PolyCam\ T_{ref})) + 18.2180 \qquad (1)$$

By design all OCAMS panchromatic filters have identical bandpasses. This allows us to use Bennu as a proxy to extend PolyCam's absolute calibration to the MapCam. To do this, we use a pair of PolyCam and MapCam images taken on 25 November 2018. Between the two images Bennu rotated one full rotation plus 1 min and 47 s (2.5 degrees). As a result, Bennu presents essentially the same face to the cameras in both images. We estimate how much this difference could affect our calibration, by comparing the integrated flux in PolyCam images taken 7 min before and 7 min after the one used to compare with MapCam. The integrated flux difference between those images is approximately 0.5%.

PolyCam and MapCam imaged Bennu at slightly different phase angles ($\Delta\alpha = 0.4$ degrees). In the time between the two images the spacecraft also closed in on Bennu by approximately 1.5 km. Correcting for these effects we estimate that the integrated flux observed by MapCam should be 1.4% greater than PolyCam's.

We use the OCAMS radiometrically calibrated frad product to integrate Bennu's flux and relate the two cameras. This product is a dark subtracted, flat fielded, radiometrically calibrated (frad = DN/s/277,035). After correcting for phase angle and distance changes, the calculated ratio between the two cameras is 24.902 and the derived calibration is given by:

$$m_R = -2.5 Log10(MapCam\_frad\_PAN \times 24.90 \times 277,035) + 18.2180 \qquad (2)$$

**Disk-integrated photometry modeling.** The ground-based campaigns covered a range of phase angles from 15.0 to 95.6 degrees yielding an absolute magnitude ($H_v$) of 20.61 ± 0.20 and phase slope ($B_v$) of 0.040 ± 0.003 magnitude per degree of phase angle. We applied a known correlation between the slope of the linear phase function and the albedo of asteroids[26,45] to estimate a global average geometric albedo of 0.030–0.045[18] for Bennu. For the spacecraft phase function campaign, we acquired images daily between 2 October and 2 December 2018. These observations yielded photometric measurements covering a phase angle range from 0.7 to 86.5 degrees in the MapCam v filter (Supplementary Table 2).

The disk-integrated Lommel-Seeliger phase function model (with an exponential phase function and a polynomial in the exponent)[27] is

$$\Phi(\alpha) = p\left[1 + \sin\frac{\alpha}{2}\tan\frac{\alpha}{2}\ln\left(\tan\frac{\alpha}{4}\right)\right]f(\alpha) \qquad (3)$$

and

$$f(\alpha) = \exp(p_1\alpha + p_2\alpha^2 + p_3\alpha^3) \qquad (4)$$

where $\alpha$ is phase angle in degrees; $p$ is geometric albedo, and $p_1$, $p_2$, and $p_3$ are parameters that defines the shape of the phase function. Resulting parameters for the Lommel-Seeliger, as well as the IAU $H,G$, Muinonen $H,G_1,G_2$ and revised $H,G_{12}$ models are given in Supplementary Table 2.

We fitted the v-band phase function data of Bennu with both the original and the revised $H, G_{12}$ models[29,30] We used the implementation of both $H, G_{12}$ models in the photometry module of the Python package for small-body planetary astronomy sbpy that is currently under development[46]. The non-linear fitting was performed with the Levenberg-Marquardt algorithm[47] as implemented in the fitting module in astropy, which is a community-developed core Python package for astronomy[48].

**Rotation rate of Bennu.** We obtained photometric measurements over two full asteroid rotations (around 8.6 h). We used the integrated flux from MapCam images by adding up the radiance from all of the pixels on Bennu to compute a lightcurve. We then compared these lightcurves with the predicted brightness using version 13 of the asteroid shape model[35] (Fig. 4).

To compute the rotational acceleration, we followed the procedure from[34], adding the data from these observations to the ground-based and Hubble Space Telescope observations from 1999, 2005, and 2012 as used in that work. We used the shape model from[35] along with the rotation pole from[49] and the observing geometry to compute synthetic lightcurve points using a Lommel-Seeliger photometric function for the observing times of the data. At each of the observation epochs, we adjusted the rotation phase of the model slightly to minimize chi-squared using the method of[50] for the data taken at that epoch. We took the absolute phase uncertainty at each of those epochs to be the amount of rotation required to increase reduced chi-squared by 1. The phase uncertainties at each epoch (3.2,1.6, 8.0, and 1.8 degrees, respectively) are slightly larger than those reported in[34], probably because the shape model used in that analysis was determined in part from those same lightcurve data, while this work uses a shape

model from spacecraft imagery. There was no 2018 data point in the earlier analysis, as these data were not yet available.

Because the absolute rotation phase is known to within 10 degrees from[34], there is no ambiguity in the absolute rotation phase, and we were able to fit a quadratic polynomial to the measured rotation phase as a function of observation time (Eq. 5).

$$P = W_0 + W_1T + W_2T^2 \qquad (5)$$

Since rotation rate is the time derivative of phase, $W_1$ is the rotation rate at $T = 0$ and $2W_2$ is the rate of change of the rotation rate. In Fig. 3, we plot this curve using $T = 0$ at the time of the first ground-based observation on 20 September 1999.

**Code availability.** This paper was produced using a number of different software packages. In some cases, versions of publicly available software were used with no custom modifications. This includes the software used for photometric reductions and manually inspection of images for dust and satellites (IRAF, http://iraf.noao.edu/ and ds9, http://ds9.si.edu/site/Home.html).

We modified a version of the Comet Toolbox code (https://bitbucket.org/Once/comet_toolbox) to model potential dust trails from Bennu[37]. Dust mass and production rate spreadsheets are a straightforward implementation of the equations in[38]. Versions of the dust trail, dust mass and production rates software and spreadsheets are available upon request to editors and reviewers.

The moving object detection software used by the Catalina Sky Survey is proprietary[19]. Two other methods were used to inspect images for satellites and other moving objects that involved no custom software and replicated and exceeded the capabilities of the Catalina Sky Survey software. The visual inspection of blinked images method defined the lower size limit of detectable satellites (Supplementary Table 1). The visual methods used the following two publicly available software packages: (IRAF, http://iraf.noao.edu/ and ds9, http://ds9.si.edu/site/Home.html).

The Bennu photometry was fitted with the Lommel-Seeliger, IAU $H,G$, and both the original[29] and the revised $H,G_{12}$ phase function models[30]. We used the implementation of the phase function models in the photometry module of the Python package for small-body planetary astronomy (sbpy) that is currently under development (https://github.com/NASA-Planetary-Science/sbpy)[46]. The non-linear fitting of the revised $H,G_{12}$ model was performed with the Levenberg-Marquardt algorithm[47] as implemented in the fitting module of astropy, which is a community-developed core Python package for astronomy[48].

## Data availability

Raw through calibrated OCAMS images will be available via the Planetary Data System (PDS) (https://sbn.psi.edu/pds/resource/orex/). Data are delivered to the PDS according to the OSIRIS-REx Data Management Plan available in the OSIRIS-REx PDS archive. Higher-level products, such as co-added dust images, Bennu photometry, and phase function model solutions, will be available in the Planetary Data System one year after departure from the asteroid. The source data underlying Figs. 3 and 4 are provided as a Source Data file. Source data for Fig. 2 is also contained in Supplementary Table 1 and for Fig. 3 in Supplementary Table 2.

The AAVSO Standard Field photometry for star cluster NGC 3532 can be found at https://www.aavso.org/apps/vsp/photometry/?east = right&fov = 30.0&scale = E&north = down&orientation = visual&maglimit = 16.5&resolution = 150&ra = 11%3A05%3A12.00&Rc = on&dec = -58%3A44%3A01.0&type=photometry&special=std_field&std_field=on. Calculations of distances and phase angles needed for mass loss rate determination is available to the public via JPL HORIZONS (https://ssd.jpl.nasa.gov/?horizons). Phase darkening coefficients for comments are available as of January 2019 from http://asteroid.lowell.edu/comet/dustphase.html.

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

## Acknowledgements

This material is based upon work supported by NASA under Contract NNM10AA11C issued through the New Frontiers Program. This work made use of sbpy (http://sbpy.org), a community-driven Python package for small-body planetary astronomy supported by NASA PDART Grant No. 80NSSC18K0987. A portion of this research was carried out at the Jet Propulsion Laboratory, California Institute of Technology, under a contract with the National Aeronautics and Space Administration. M.A.B. and S.F. acknowledge financial support from CNES. We thank Vishnu Reddy for useful suggestions to improve the manuscript.

## Author contributions

C.W.H. is the head of the OSIRIS-REx Astronomy Working Group. He is responsible for the design and implementation of the Approach phase campaign and led the data reduction. C.K.M. is a member of the Astronomy Working Group and responsible for Finson-Probstein analysis and estimating bounds on the dust trail mass loss rate. M.C.N. provided the lightcurve analysis and text for the manuscript in this section. J-Y.L., B.E.C., and X-D.Z. provided text and fitting of photometric models to our phase function data. C.D.d'A., D.N.D, D.R.G. and P.H.S. are responsible for calibration of the OCAMS instrument and related data reduction tasks. F.C.S. and E.J.C. analyzed the hazard search images for detections of natural satellites. T.R.K. assisted in data reduction of the phase function photometry. S.F., M.A.B., H.C., and M.R.M.I., provided insight and contributed text into comparing our results from Bennu to other spacecraft targets such as 67P and Ryugu. S.R.C. and D.J.S. were involved in the design of the hazards search. E.B.B., C.M. H., E.S.H., B.R. provided commentary and suggestions to the manuscript. D.S.L leads the OSIRIS-REx mission and is responsible for the overall scientific investigation, the development and implementation of mission requirements, and leadership of the science and operations mission teams. He contributed substantially to the content of this manuscript. The OSIRIS-REx team is in charge of executing the OSIRIS-REx mission for NASA, and is responsible for producing data products from mission instruments that will be made available to the scientific community.

## Additional information

**Competing interests:** The authors declare no competing interests.

## The OSIRIS-REx Team

D.E. Highsmith[11], J. Small[11], D. Vokrouhlický[12], N.E. Bowles[13], E. Brown[13], K.L. Donaldson Hanna[13], T. Warren[13], C. Brunet[14], R.A. Chicoine[14], S. Desjardins[14], D. Gaudreau[14], T. Haltigin[14], S. Millington-Veloza[14], A. Rubi[14], J. Aponte[15], N. Gorius[15], A. Lunsford[15], B. Allen[16], J. Grindlay[16], D. Guevel[16], D. Hoak[16], J. Hong[16], D.L. Schrader[17], J. Bayron[18], O. Golubov[19], P. Sánchez[19], J. Stromberg[20], M. Hirabayashi[21], C.M. Hartzell[9], S. Oliver[22], M. Rascon[22], A. Harch[23], J. Joseph[23], S. Squyres[23], D. Richardson[24], J.P. Emery[25], L. McGraw[25], R. Ghent[26], R.P. Binzel[27], M.M. Al Asad[28], C.L. Johnson[2,28], L. Philpott[28], H.C.M. Susorney[28], E.A. Cloutis[29], R.D. Hanna[30], H.C. Connolly Jr.[31], F. Ciceri[32], A.R. Hildebrand[32], E.-M. Ibrahim[32], L. Breitenfeld[33], T. Glotch[33], A.D. Rogers[33], B.E. Clark[8], S. Ferrone[8], C.A. Thomas[34], H. Campins[6], Y. Fernandez[6], W. Chang[35], A. Cheuvront[36], D. Trang[37], S. Tachibana[38], H. Yurimoto[38], J.R. Brucato[39], G. Poggiali[39], M. Pajola[40], E. Dotto[41], E. Mazzotta Epifani[41], M.K. Crombie[42], C. Lantz[43], M.R.M. Izawa[44], J. de Leon[44], J. Licandro[44], J.L.Rizos Garcia[44], S. Clemett[45], K. Thomas-Keprta[45], S. Van wal[46], M. Yoshikawa[46], J. Bellerose[7], S. Bhaskaran[7], C. Boyles[7], S.R. Chesley[7], C.M. Elder[7], D. Farnocchia[7], A. Harbison[7], B. Kennedy[7], A. Knight[7], N. Martinez-Vlasoff[7], N. Mastrodemos[7], T. McElrath[7], W. Owen[7], R. Park[7], B. Rush[7], L. Swanson[7], Y. Takahashi[7], D. Velez[7], K. Yetter[7], C. Thayer[47], C. Adam[48], P. Antreasian[48], J. Bauman[48], C. Bryan[48], B. Carcich[48], M. Corvin[48], J. Geeraert[48], J. Hoffman[48], J.M. Leonard[48], E. Lessac-Chenen[48], A. Levine[48], J. McAdams[48], L. McCarthy[48], D. Nelson[48], B. Page[48], J. Pelgrift[48], E. Sahr[48], K. Stakkestad[48], D. Stanbridge[48], D. Wibben[48], B. Williams[48], K. Williams[48], P. Wolff[48], P. Hayne[49], D. Kubitschek[49], M.A. Barucci[4], J.D.P. Deshapriya[4], S. Fornasier[4], M. Fulchignoni[4], P. Hasselmann[4], F. Merlin[4], A. Praet[4], E.B. Bierhaus[5], O. Billett[5], A. Boggs[5], B. Buck[5], S. Carlson-Kelly[5], J. Cerna[5], K. Chaffin[5], E. Church[5], M. Coltrin[5], J. Daly[5], A. Deguzman[5], R. Dubisher[5], D. Eckart[5], D. Ellis[5], P. Falkenstern[5], A. Fisher[5], M.E. Fisher[5], P. Fleming[5], K. Fortney[5], S. Francis[5], S. Freund[5], S. Gonzales[5], P. Haas[5], A. Hasten[5], D. Hauf[5], A. Hilbert[5], D. Howell[5], F. Jaen[5], N. Jayakody[5], M. Jenkins[5], K. Johnson[5], M. Lefevre[5], H. Ma[5], C. Mario[5], K. Martin[5], C. May[5], M. McGee[5], B. Miller[5], C. Miller[5], G. Miller[5], A. Mirfakhrai[5], E. Muhle[5], C. Norman[5], R. Olds[5], C. Parish[5], M. Ryle[5], M. Schmitzer[5], P. Sherman[5], M. Skeen[5], M. Susak[5], B. Sutter[5], Q. Tran[5], C. Welch[5], R. Witherspoon[5], J. Wood[5], J. Zareski[5], M. Arvizu-Jakubicki[1], E. Asphaug[1], E. Audi[1], R.-L. Ballouz[1], R. Bandrowski[1], K.J. Becker[1], T.L. Becker[1], S. Bendall[1], C.A. Bennett[1], H. Bloomenthal[1], D. Blum[1], W.V. Boynton[1], J. Brodbeck[1], K.N. Burke[1], M. Chojnacki[1], A. Colpo[1], J. Contreras[1], J. Cutts[1], C.Y. Drouet d'Aubigny[1], D. Dean[1], D.N. DellaGiustina[1], B. Diallo[1], D. Drinnon[1], K. Drozd[1], H.L. Enos[1], R. Enos[1], C. Fellows[1], T. Ferro[1], M.R. Fisher[1], G. Fitzgibbon[1], M. Fitzgibbon[1], J. Forelli[1], T. Forrester[1], I. Galinsky[1], R. Garcia[1], A. Gardner[1], D.R. Golish[1], N. Habib[1], D. Hamara[1], D. Hammond[1], K. Hanley[1], K. Harshman[1], C.W. Hergenrother[1], K. Herzog[1], D. Hill[1], C. Hoekenga[1], S. Hooven[1], E.S. Howell[1], E. Huettner[1], A. Janakus[1], J. Jones[1], T.R. Kareta[1], J. Kidd[1], K. Kingsbury[1], S.S. Balram-Knutson[1], L. Koelbel[1], J. Kreiner[1], D. Lambert[1], D.S. Lauretta[1], C. Lewin[1], B. Lovelace[1], M. Loveridge[1], M. Lujan[1], C.K. Maleszewski[1], R. Malhotra[1], K. Marchese[1], E. McDonough[1], N. Mogk[1], V. Morrison[1], E. Morton[1],

R. Munoz[1], J. Nelson[1], M.C. Nolan[1], J. Padilla[1], R. Pennington[1], A. Polit[1], N. Ramos[1], V. Reddy[1], M. Riehl[1], B. Rizk[1], H.L. Roper[1], S. Salazar[1], S.R. Schwartz[1], S. Selznick[1], N. Shultz[1], P.H. Smith[1], S. Stewart[1], S. Sutton[1], T. Swindle[1], Y.H. Tang[1], M. Westermann[1], C.W.V. Wolner[1], D. Worden[1], T. Zega[1], Z. Zeszut[1], A. Bjurstrom[50], L. Bloomquist[50], C. Dickinson[50], E. Keates[50], J. Liang[50], V. Nifo[50], A. Taylor[50], F. Teti[50], M. Caplinger[51], H. Bowles[52], S. Carter[52], S. Dickenshied[52], D. Doerres[52], T. Fisher[52], W. Hagee[52], J. Hill[52], M. Miner[52], D. Noss[52], N. Piacentine[52], M. Smith[52], A. Toland[52], P. Wren[52], M. Bernacki[53], D. Pino Munoz[53], S.-i. Watanabe[46,54], S.A. Sandford[55], A. Aqueche[56], B. Ashman[56], M. Barker[56], A. Bartels[56], K. Berry[56], B. Bos[56], R. Burns[56], A. Calloway[56], R. Carpenter[56], N. Castro[56], R. Cosentino[56], J. Donaldson[56], J.P. Dworkin[56], J. Elsila Cook[56], C. Emr[56], D. Everett[56], D. Fennell[56], K. Fleshman[56], D. Folta[56], D. Gallagher[56], J. Garvin[56], K. Getzandanner[56], D. Glavin[56], S. Hull[56], K. Hyde[56], H. Ido[56], A. Ingegneri[56], N. Jones[56], P. Kaotira[56], L.F. Lim[56], A. Liounis[56], C. Lorentson[56], D. Lorenz[56], J. Lyzhoft[56], E.M. Mazarico[56], R. Mink[56], W. Moore[56], M. Moreau[56], S. Mullen[56], J. Nagy[56], G. Neumann[56], J. Nuth[56], D. Poland[56], D.C. Reuter[56], L. Rhoads[56], S. Rieger[56], D. Rowlands[56], D. Sallitt[56], A. Scroggins[56], G. Shaw[56], A.A. Simon[56], J. Swenson[56], P. Vasudeva[56], M. Wasser[56], R. Zellar[56], J. Grossman[57], G. Johnston[57], M. Morris[57], J. Wendel[57], A. Burton[58], L.P. Keller[58], L. McNamara[58], S. Messenger[58], K. Nakamura-Messenger[58], A. Nguyen[58], K. Righter[58], E. Queen[59], K. Bellamy[60], K. Dill[60], S. Gardner[60], M. Giuntini[60], B. Key[60], J. Kissell[60], D. Patterson[60], D. Vaughan[60], B. Wright[60], R.W. Gaskell[2], L. Le Corre[2], J.-Y. Li[2], J.L. Molaro[2], E.E. Palmer[2], M.A. Siegler[2], P. Tricarico[2], J.R. Weirich[2], X.-D. Zou[2], T. Ireland[61], K. Tait[62], P. Bland[63], S. Anwar[64], N. Bojorquez-Murphy[64], P.R. Christensen[64], C.W. Haberle[64], G. Mehall[64], K. Rios[64], I. Franchi[65], B. Rozitis[65], C.B. Beddingfield[66], J. Marshall[66], D.N. Brack[10], A.S. French[10], J.W. McMahon[10], D.J. Scheeres[10], E.R. Jawin[67], T.J. McCoy[67], S. Russell[67], M. Killgore[68], W.F. Bottke[69], V.E. Hamilton[69], H.H. Kaplan[69], K.J. Walsh[69], J.L. Bandfield[70], B.C. Clark[70], M. Chodas[71], M. Lambert[71], R.A. Masterson[71], M.G. Daly[72], J. Freemantle[72], J.A. Seabrook[72], O.S. Barnouin[73], K. Craft[73], R.T. Daly[73], C. Ernst[73], R.C. Espiritu[73], M. Holdridge[73], M. Jones[73], A.H. Nair[73], L. Nguyen[73], J. Peachey[73], M.E. Perry[73], J. Plescia[73], J.H. Roberts[73], R. Steele[73], R. Turner[73], J. Backer[74], K. Edmundson[74], J. Mapel[74], M. Milazzo[74], S. Sides[74], C. Manzoni[75], B. May[75], M. Delbo'[76], G. Libourel[76], P. Michel[76], A. Ryan[76], F. Thuillet[76] & B. Marty[77]

[11]Aerospace Corporation, Chantilly, VA, USA. [12]Astronomical Institute, Charles University, Prague, Czech Republic. [13]Atmospheric, Oceanic and Planetary Physics, University of Oxford, Oxford, UK. [14]Canadian Space Agency, Saint-Hubert, Quebec, Canada. [15]Catholic University of America, Washington, DC, USA. [16]Center for Astrophysics, Harvard University, Cambridge, MA, USA. [17]Center for Meteorite Studies, Arizona State University, Tempe, AZ, USA. [18]City University of New York, New York, NY, USA. [19]Colorado Center for Astrodynamics Research, University of Colorado, Boulder, CO, USA. [20]Commonwealth Scientific and Industrial Research Organisation (CSIRO), Canberra, Australian Capital Territory, Australia. [21]Department of Aerospace Engineering, Auburn University, Auburn, AL, USA. [22]Department of Astronomy and Steward Observatory, University of Arizona, Tuscon, AZ, USA. [23]Department of Astronomy, Cornell University, Ithaca, NY, USA. [24]Department of Astronomy, University of Maryland, College Park, MD, USA. [25]Department of Earth and Planetary Sciences, University of Tennessee, Knoxville, TN, USA. [26]Department of Earth Sciences, University of Toronto, Toronto, Ontario, Canada. [27]Department of Earth, Atmospheric, and Planetary Sciences, Massachusetts Institute of Technology, Cambridge, MA, USA. [28]Present address: Department of Earth, Ocean and Atmospheric Sciences, University of British Columbia, Vancouver, British Columbia, Canada. [29]Department of Geography, University of Winnipeg, Winnipeg, Manitoba, Canada. [30]Department of Geological Sciences, Jackson School of Geosciences, University of Texas, Austin, TX, USA. [31]Department of Geology, Rowan University, Glassboro, NJ, USA. [32]Department of Geoscience, University of Calgary, Calgary, Alberta, Canada. [33]Department of Geosciences, Stony Brook University, Stony Brook, NY, USA. [34]Department of Physics and Astronomy, Northern Arizona University, Flagstaff, AZ, USA. [35]Edge Space Systems, Greenbelt, MD, USA. [36]General Dynamics C4 Systems, Denver, CO, USA. [37]Hawai'i Institute of Geophysics and Planetology, University of Hawai'i at Mānoa, Honolulu, HI, USA. [38]Hokkaido University, Sapporo, Japan. [39]INAF–Astrophysical Observatory of Arcetri, Florence, Italy. [40]INAF–Osservatorio Astronomico di Padova, Padova, Italy. [41]INAF–Osservatorio Astronomico di Roma, Rome, Italy. [42]Indigo Information Services, Tucson, AZ, USA. [43]Institut d'Astrophysique Spatiale, CNRS/Université Paris Sud, Orsay, France. [44]Instituto de Astrofísica de Canarias and Departamento de Astrofísica, Universidad de La Laguna, Tenerife, Spain. [45]Jacobs Technology, Houston, TX, USA. [46]JAXA Institute of Space and Astronautical Science, Sagamihara, Japan. [47]Kavli Institute for Astrophysics and Space Research, Massachusetts Institute of Technology, Cambridge, MA, USA. [48]KinetX Aerospace, Inc., Simi Valley, CA, USA. [49]Laboratory for Atmospheric and Space Physics, University of Colorado, Boulder, CO, USA. [50]Macdonald, Dettwiler, and Associates, Brampton, Ontario, Canada. [51]Malin Space Science Systems, San Diego, CA, USA. [52]Mars Space Flight Facility, Arizona State University, Tempe, AZ, USA. [53]Mines ParisTech, Paris, France. [54]Nagoya University, Nagoya, Japan. [55]NASA Ames Research Center, Moffett Field, CA, USA. [56]NASA Goddard Space Flight Center, Greenbelt, MD, USA. [57]NASA Headquarters, Washington, DC, USA. [58]NASA Johnson Space Center, Houston, TX, USA. [59]NASA Langley Research Center, Hampton, VA, USA. [60]NASA Marshall Space Flight Center, Huntsville, AL, USA. [61]Research School of Earth Sciences, Australian National University, Canberra, Australian Capital Territory, Australia. [62]Royal Ontario Museum, Toronto, Ontario, Canada. [63]School of Earth and Planetary Sciences, Curtin University, Perth, Wstern Australia, Australia. [64]School of Earth and Space Exploration, Arizona State University, Tempe, AZ, USA. [65]School of Physical Sciences, The Open

University, Milton Keynes, UK. [66]SETI Institute, Mountain View, CA, USA. [67]Smithsonian Institution National Museum of Natural History, Washington, DC, USA. [68]Southwest Meteorite Laboratory, Payson, AZ, USA. [69]Southwest Research Institute, Boulder, CO, USA. [70]Space Science Institute, Boulder, CO, USA. [71]Space Systems Laboratory, Department of Aeronautics and Astronautics, Massachusetts Institute of Technology, Cambridge, MA, USA. [72]The Centre for Research in Earth and Space Science, York University, Toronto, Ontario, Canada. [73]The Johns Hopkins University Applied Physics Laboratory, Laurel, MD, USA. [74]U.S. Geological Survey Astrogeology Science Center, Flagstaff, AZ, USA. [75]London Stereoscopic Company, London, UK. [76]Université Côte d'Azur, Observatoire de la Côte d'Azur, CNRS, Laboratoire Lagrange, Nice, France. [77]Université de Lorraine, Nancy, France

