## [Peer Review File · Nature Communications]

Reviewers' comments:

Reviewer #1 (Remarks to the Author):

General remarks: The paper is very interesting and useful for scientists investigating physical properties, both the asteroid Bennu and other space bodies that were studied from space missions. The all assumptions are spelled out clearly, the methods are described comprehensively and the figures and tables are necessary and properly laid out. I have carefully read the manuscript and found no substantive comments that would not be allowed to publish it. The paper satisfies a subject area of the Journal and may be published after minor revisions.

REMARKS

Lines 23-24: You would not write full name of the spacecraft in the abstract. It is better to give the full name at the beginning of the paper.

Line 33: Degree in acceleration should be as a superscript.

Line 132: Reference [24] does not relate to the Lommel-Seeliger model. See remark to line 588. It should be corrected.

Lines 132-137: Pentilla et al. (Pentilla et al. 2016. Planet. Space Sci. 123, 117-125) using a new data set of high-quality magnitude-phase functions of 93 asteroids (Shevchenko et al. 2016. Planet. Space Sci. 123, 101-116) re-assessed the two-parameter version of the HG1G2 function and introduced a one-parameter version of the phase function that can give a suggestion of the asteroids taxonomic group based only on its phase curve. An online tool that implements these algorithms is introduced on site: <http://www.helsinki.fi/project/psr/HG1G2/>. Please, recalculate your disk-integrated data using new versions of these functions and insert obtained new parameters in the Supplementary Table 2.

Lines 150-156: Shevchenko & Belskaya (Shevchenko & Belskaya 2010, EPSC Abstracts. Vol. 5, 738) analyzed magnitude-phase functions of 33 dark asteroids pointed out that about 40% of dark asteroids displayed wide opposition effect starting at the phase angle 6-7 deg with amplitude of 0.15- 0.25 mag. Please, take into account this in your analysis.

Line 154: Reference [24] should be changed on [21].

Line 165: Reference [15] should be changed on [13].

Line 168: Reference [14] should be changed on [13].

Line 404: Reference [23] should be changed on [27].

Line 588: There is no reference on Hapke 2012 in the list of references.

Line 601: It is needed to insert a number of reference Lauretta 2019.

Please, insert of your disk-integrated data on the magnitude-phase relation of Bennu in supplementary Table for more investigation.

Vasilij Shevchenko

Reviewer #2 (Remarks to the Author):

This paper treats a quite original data and must be published. It is useful information for the research communities that each result shows the good agreement between the ground-based observations/other mission data/several models and the OSIRIS-Rex data.

Comments:

- Search for Natural Satellites (P.2): The paper indicates the detectable size and the area range but it seems not to mention the possibility of the moving objects in front of Bennu or at the back of Bennu during the observations; Due to the long-term observation dates, the orbital motion would, of course, reveal the existence of the objects.
- Supplementary Table 1: Phase angle, Distance from Bennu searched etc. have only one value at each date. Didn't they change during the "start and end range of search"? Especially, the phase angle is correct to three significant figures.

Minor comments:

- The expression of dates should be uniformed: 12 September 2018 / 2018 Nov. 2 / 2018-Oct-23.
- You use "approach" with capital "A", such as "Approach phase." A similar expression, for example, is "Detailed Survey phase." Do the capital words have special meanings in the mission? If so, it is ok.
- Line 371(Figure 2): 0.24m is expressed as 24 cm in other sections.
- Line 389/390(Figure 3): The red points is blue? Also black line is purple broken line? My PC display/printed papers show different colors from the captions.
- Line 439: AAM2 should be explained the meaning (the meaning of AAM3 is at Line 515.)
- Line 480: distances is distance?
- Line 490: AU is au.
- Line 497: Do we need "from"?
- Supplementary Table 2: Each column should insert each unit if they need.

Reviewer #3 (Remarks to the Author):

Paper presents solid but not particularly interesting results on Bennu. I do not see anything wrong with this paper but am unsure about its suitability for the journal.

Response to reviewer's comments for manuscript NCOMMS-19-03341 "The Operational Environment and Rotational Acceleration of Asteroid (101955) Bennu from OSIRIS-REx Observations"

We addressed all remarks made by 2 of the 3 reviewers. The third reviewer made no actionable remarks. Our responses are given in green below each remark. Some remarks with similar responses are grouped together.

Reviewer #1 (Remarks to the Author) (Vasilij Shevchenko):

1. Lines 23-24: You would not write full name of the spacecraft in the abstract. It is better to give the full name at the beginning of the paper.

We defer to the editor's discretion as to whether the full name of the spacecraft is given in the abstract or the introduction. Currently, the full name is given in both locations.

2. Line 33: Degree in acceleration should be as a superscript.

Degree changed from normal script word (deg) to superscript symbol ($^{\circ}$).

3. Line 132: Reference [24] does not relate to the Lommel-Seeliger model. See remark to line 588. It should be corrected.

Lommel-Seeliger reference changed to Hapke (2012).

4. Lines 132-137: Penttilla et al. (Penttilla et al. 2016. Planet. Space Sci. 123, 117-125) using a new data set of high-quality magnitude-phase functions of 93 asteroids (Shevchenko et al. 2016. Planet. Space Sci. 123, 101-116) re-assessed the two-parameter version of the HG1G2 function and introduced a one-parameter version of the phase function that can give a suggestion of the asteroids taxonomic group based only on its phase curve. An online tool that implements these algorithms is introduced on site: <http://www.helsinki.fi/project/psr/HG1G2/>. Please, recalculate your disk-integrated data using new versions of these functions and insert obtained new parameters in the Supplementary Table 2.

We recomputed the H,G12 phase function using the implementation of Penttilla et al. (2016). The change was minor relative to the values given by the original H,G12 implementation in Muinonen et al. (2010). All H,G12 parameter values in the main text, Figure 3, and Supplementary Table 2 have been changed to the Penttilla et al. (2016) values.

5. Lines 150-156: Shevchenko & Belskaya (Shevchenko & Belskaya 2010, EPSC Abstracts. Vol. 5, 738) analyzed magnitude-phase functions of 33 dark asteroids pointed out that about 40% of dark asteroids displayed wide opposition effect starting at the phase angle 6-7 deg with amplitude of 0.15- 0.25 mag. Please, take into account this in your analysis.

We agree with the reviewer's comment and have modified the text to state that the broad opposition effect of Bennu is displayed by ~40% of dark asteroids.

6. Line 154: Reference [24] should be changed on [21].

7. Line 165: Reference [15] should be changed on [13].

8. Line 168: Reference [14] should be changed on [13].

9. Line 404: Reference [23] should be changed on [27].

Reference numbers changed, though many of our reference numbers have changed again as references have been added and text moved in this revision.

10. Line 588: There is no reference on Hapke 2012 in the list of references.

Reference for Hapke 2012 has been added to the reference list.

11. Line 601: It is needed to insert a number of reference Laurretta 2019.
Reference number for Laurretta 2019 added to text.

12. Please, insert of your disk-integrated data on the magnitude-phase relation of Bennu in supplementary Table for more investigation.

A new Supplementary Table has been added containing the Bennu photometry used to model its phase function. This data is also contained in the Source Data File.

Reviewer #2

Comments:

13. Search for Natural Satellites (P.2): The paper indicates the detectable size and the area range but it seems not to mention the possibility of the moving objects in front of Bennu or at the back of Bennu during the observations; Due to the long-term observation dates, the orbital motion would, of course, reveal the existence of the objects.

We now acknowledge that some satellite orbits result in the objects being located in front of or behind Bennu for a fraction of their orbits. As the reviewer states, the multiple search dates and 5-hour observing windows per date ensured that orbiting objects were detectable away from the interference of Bennu.

14. Supplementary Table 1: Phase angle, Distance from Bennu searched etc. have only one value at each date. Didn't they change during the "start and end range of search"? Especially, the phase angle is correct to three significant figures.

The caption now explains that the phase angle, distance from Bennu searched, and smallest detectable diameters are valid for the end of the daily observations which occurred at 9:00 UTC every day.

Minor comments:

15. The expression of dates should be uniformed: 12 September 2018 / 2018 Nov. 2 / 2018-Oct-23.

We have changed all dates to be uniform using the following format: " DD MONTH YYYY ".

16. You use "approach" with capital "A", such as "Approach phase." A similar expression, for example, is "Detailed Survey phase." Do the captial words have special meanings in the mission? If so, it is ok.

The mission phases during the encounter with Bennu are capitalized by the OSIRIS-REx mission. Unless the editors object, we ask that Approach and Detailed Survey remain capitalized as per OSIRIS-REx usage.

17. Line 371(Figure 2): 0.24m is expressed as 24 cm in other sections.

0.24m has been changed to 24 cm in the caption for Figure 2.

18. Line 389/390(Figure 3): The red points is blue? Also black line is purple broken line? My PC display/printed papers show different colors from the captions.

The reviewer is correct. We changed the caption of Figure 3 to describe the correct colors ...

19. Line 439: AAM2 should be explained the meaning (the meaning of AAM3 is at Line 515.)

The reference to AAM2 has been expanded to explain its meaning.

20. Line 480: distances is distance?

21. Line 490: AU is au.

22. Line 497: Do we need "from"?

Typo corrected.

23. Supplementary Table 2: Each column should insert each unit if they need.

The caption states that the wavelength column is in nanometers and the other columns are unitless.

Reviewer #3:

This reviewer made no actionable remarks.

REVIEWERS' COMMENTS:

Reviewer #1 (Remarks to the Author):

I am satisfied the answers of the authors and I have no other remarks. The paper can be published in the journal.

Reviewer #2 (Remarks to the Author):

The authors' corrections are sufficient for my comments.